# CRISPR FokI Dead Cas9 System: Principles and Applications in Genome Engineering

**DOI:** 10.3390/cells9112518

**Published:** 2020-11-21

**Authors:** Maryam Saifaldeen, Dana E. Al-Ansari, Dindial Ramotar, Mustapha Aouida

**Affiliations:** College of Health and Life Sciences, Division of Biological and Biomedical Sciences, Hamad Bin Khalifa University, Education City, Qatar Foundation, Doha P.O.Box 34110, Qatar; msaifaldeen@hbku.edu.qa (M.S.); DAlAnsari@hbku.edu.qa (D.E.A.-A.)

**Keywords:** genome engineering, CRISPR, dead Cas9, FokI endonuclease, FokI-dCas9, RNA-guided FokI nuclease, spacer sequence, dual guide RNA, fdCas9 variants

## Abstract

The identification of the robust clustered regularly interspersed short palindromic repeats (CRISPR) associated endonuclease (Cas9) system gene-editing tool has opened up a wide range of potential therapeutic applications that were restricted by more complex tools, including zinc finger nucleases (ZFNs) and transcription activator-like effector nucleases (TALENs). Nevertheless, the high frequency of CRISPR system off-target activity still limits its applications, and, thus, advanced strategies for highly specific CRISPR/Cas9-mediated genome editing are continuously under development including CRISPR–FokI dead Cas9 (fdCas9). fdCas9 system is derived from linking a FokI endonuclease catalytic domain to an inactive Cas9 protein and requires a pair of guide sgRNAs that bind to the sense and antisense strands of the DNA in a protospacer adjacent motif (PAM)-out orientation, with a defined spacer sequence range around the target site. The dimerization of FokI domains generates DNA double-strand breaks, which activates the DNA repair machinery and results in genomic edit. So far, all the engineered fdCas9 variants have shown promising gene-editing activities in human cells when compared to other platforms. Herein, we review the advantages of all published variants of fdCas9 and their current applications in genome engineering.

## 1. Introduction

Genome engineering has been applied to different types of cells and organisms and is used as an essential tool in biological and biomedical research [1]. Through genome engineering tools, a defective chromosomal sequence can be altered by inducing a double-stranded break (DSB), which stimulates the DNA repair pathway and results in gene expression manipulation [2]. There are two main DNA repair pathways stimulated by DNA editing tools: (i) homology-directed repair (HDR) and (ii) canonical nonhomologous end-joining (c-NHEJ) [2]. HDR relies on the delivery of a corrective donor DNA template, which is flanked by complementary overhang homology arms to allow gene replacement using the cellular homologous recombination process [3]. HDR mechanism is considered to have lower gene-editing efficiency but is favored for precise gene knock-in and knockout applications [4,5]. In contrast, the c-NHEJ results in the addition or deletion of short nucleotide sequences at the site of the DSB, causing a reading frame shift, which makes it ideal for the gain/loss of gene function applications [6,7,8]. Generally, the trend of favoring c-NHEJ machinery over HDR is common to most gene-editing tools because it occurs independent of the cell cycle stage, unlike HDR, which is limited to the S/G2 phase [3]. In most cases, the decreased rates of precise homologous recombination by HDR can be enhanced through NHEJ pathway inhibitors or HDR enhancer drugs [9,10].

The three commonly used nuclease-based genome engineering tools are (i) the zinc finger nucleases (ZFNs), (ii) the transcription activator-like effector nucleases (TALENs), and (iii) the clustered regularly interspersed short palindromic repeats (CRISPR) associated endonuclease (Cas9) system [11,12]. These nuclease-based systems can create undesired off-target effects, and, therefore, identifying new genome engineering tools that minimize these effects are needed [4,13,14,15,16,17,18,19,20,21]. In 2014, a new derivative of the CRISPR genome engineering tool was introduced, where an inactive “dead” form of Cas9 (dCas9) is fused to the FokI endonuclease catalytic domain, referred to as FokI–dCas9 (fdCas9) [22,23,24,25,26]. In this review, we will describe the principles and limitations of all the abovementioned gene engineering tools (ZFNs, TALENs, CRISPR–Cas9), compare them to fdCas9 (Figure 1, Table 1 and Table 2), and discuss the engineering of the different variants of the chimeric fdCas9 (Figure 2 and Table 3) [22,23,24,25,26]. Finally, we will conclude with the current applications that validate the efficiency of using fdCas9 in genome engineering and some of the upgraded properties to other existing gene-editing tools.

## 2. Gene-Editing Tools

### 2.1. Zinc Finger Nucleases (ZFNs)

Zinc finger nucleases (ZFNs) were the first genome engineering tool designed to artificially induce highly specific DSBs in targeted regions of the genome by utilizing the most commonly found DNA-binding motif, the zinc finger (ZF) [27]. ZF consists of seven conserved amino acids, Cys–Cys and His–His paired through a zinc ion, as well as three additional amino acids, Tyr6, Phe17, and Leu23, that are responsible for the characteristic arrangement of the ββα tertiary structure [28]. Each ZF motif recognizes three nucleotides through interactions with three variable amino acid residues located at Positions 2, 3, and 6 on the α helix [29]. A single ZFN molecule consists of two domains: (i) a specific DNA-binding module consisting of 3 to 4 ZF motifs, which collectively recognizes 9 to 12 bp of the targeted DNA, and (ii) a nonspecific catalytic module of the FokI nuclease [30,31]. For successful genome engineering using ZFNs, two units of ZFNs must be engineered to colocalize at two adjacent sites on opposite strands of the DNA target, with a spacer distance of 5 to 7 bp between the two sites to allow the obligate FokI dimerization (Figure 1, Table 1) [32]. Despite its specificity of recognizing 18 to 24 bp target DNA and the ease of delivery due to its small size, there are several limitations of ZFNs that hinders their use as a reliable genome-editing tool [29]. These include (i) the DNA interaction of the ZF motif that is restricted to multiples of three, (ii) the creation of undesired off-target mutations, (iii) the limited selection of the target site, which is estimated to be every 50 to 200 bps of the whole genome, and (iv) the construct design that involves complex protein engineering, which requires longer time, greater cost, and is not suitable for routine clinical use [4,33].

### 2.2. Transcription Activator-Like Effector Nucleases (TALENs)

A class of DNA-binding protein transcription activator-like effectors (TALEs) were first discovered in *Xanthomonas,* a bacterial pathogen of plants [34]. TALEs are characterized by 12–28 repeats of the DNA-binding domain, a nuclear localization signal (NLS), and an acidic transcriptional activation domain at the C-terminal region [35,36]. Each DNA-binding domain consists of 34 amino acid residues, with a polymorphism located at Positions 12 and 13, known as repeat variable di-residues (RVDs) that determine nucleotide-binding specificity (Figure 1) [37]. By fusing the catalytic FokI endonuclease domain to the C-terminus of the TALE DNA-binding domain, the genome engineering tool TALE nucleases (TALENs) emerged [37]. TALENs are distinct from ZFNs as they bind in a ratio of one RVD to one nucleotide, resulting in broader target site selection [38]. In addition to the obligate FokI dimerization requirement, TALENs have a specificity of 30 to 40 bp, with a favored spacer distance of 14 to 16 bp between the target sites (Figure 1, Table 1) [12]. Although TALENs are considered simpler, less cytotoxic, and less time-consuming to engineer than ZFNs [39], the large cDNA size (~3 Kb compared to ~1 Kb for ZFNs) limits its packaging and delivery [13,39]. It is noteworthy that TALENs have limited off-target effects when compared to ZFNs, but the cost of engineering this system is much higher [1,11].

### 2.3. CRISPR/Cas Systems

The CRISPR/Cas9 system was first identified as a part of the bacterial immune response against bacteriophages and is now repurposed as a powerful genome-engineering tool that has been successfully used across many eukaryotic species [40,41,42,43,44,45]. Generally, the system is composed of an endonuclease Cas protein that drives DNA cleavage activity and a single guide RNA molecule (sgRNA) that determines the system’s specificity through binding to the targeted gene, following the Watson–Crick base-pairing rule [41,42,45]. A variety of Cas proteins have been identified, each recognizing different protospacer adjacent motif (PAM) sequences at the target sequence [46]. In the case of the CRISPR/Cas9 system, the *Streptococcus pyogenes* Cas9 (SpCas9) endonuclease recognizes the 5′-NGG-3′ PAM sequence, where N could be any nucleotide [47]. DSBs are created through the two HNH and RuvC endonuclease domains of SpCas9, cleaving the complementary strand and the noncomplementary strand, respectively [47] (Figure 1). The sgRNA required for the activation of the CRISPR/Cas9 system is composed of a 17–20-nucleotide CRISPR RNA (crRNA) targeting sequence that is a complementary target site and the 85-nucleotide transactivating CRISPR RNA (tracrRNA) processing sequence, which provides structural stability of the ribonucleoprotein complex (RNPC) [48,49]. Upon PAM recognition, sgRNA binding, and the formation of the RNPC, the two catalytic nuclease domains induce DSB 3 to 4 nucleotides upstream of the PAM site, resulting in the activation of DNA repair machinery [50].

One of the most important advantages of the CRISPR/Cas9 system is the robustness and ease of programming the system through redesigning the sgRNA, the crRNA in particular, into new genes of interest, with the protein constructs remaining constant. In ZFNs and TALENs, complex molecular cloning methods and substantial protein engineering are required with every target design [11,12,51]. Although enhanced versions of ZFNs and TALENs have slightly improved their multiplexing efficiencies, the CRISPR system is the easiest tool to multiplex and generate simultaneous gene edits [40,52,53,54,55]. Nevertheless, there are some limitations of using the CRISPR/Cas9 system in genome engineering, which include PAM sequence availability at the target sites and, most importantly, the increased frequency of nonspecific off-target mutations reported in many studies, compared to ZFNs and TALENs (Table 1) [15,16,17,18,19].

#### 2.3.1. Cas9 Variants

##### Cas9 Nickases

SpCas9 nickases are partially inactive, with either HNH or RuvC endonuclease activity being abolished by the variants D10A and H840A, respectively [56]. These nickases can produce DNA single-strand nicks instead of DNA DSBs, which correlates to the decreased off-target mutation rates when compared to wild-type (WT) Cas9 [57]. Genome engineering mediated by Cas9 nickases is inherently more complicated than a WT Cas9 [57,58,59]. The requirements are for two sgRNAs functioning simultaneously to generate DSBs, with the predicted overhang lengths and polarities, and PAMs in an outward orientation (Table 2, column 3) [58,59]. It is noteworthy that Cas9 D10A-mediated genome engineering is more robust when the two cleavage sites are 37 to 68 bp apart, while Cas9 H840A favors a distance of 51 to 68 bp [60]. Although Cas9 nickases have shown a great decrease in known off-target genes, deep sequencing shows elevated levels of point mutations at the target site, indicating that DNA nicks are still considered mutagenic even when nickases/sgRNA complexes are expressed monomerically [19,21,22,58,61].

##### Inactive Cas9 (Dead Cas9)

Another form of spCas9 that has been introduced is the inactive dCas9, which is formed by inactivating both HNH and RuvC endonuclease domains [62]. Because of the lack of nuclease activity in the dCas9, it has been utilized in many applications that exploited its DNA detection and localization and the ability to synthesize chimeric protein complexes when fused with different effector proteins [61,62,63]. These applications include genomic visualization via the fusion with fluorescent proteins, gene regulation through fusion with activators or repressors, alteration in epigenetic modifications through fusion with methyltransferases or deacetylases, and immunoprecipitation [62,63,64].

##### Base Editors

Base editors have recently emerged as efficient and specific genome engineering tools through the fusion of dCas9 with a catalytic domain that is capable of deaminating cytosine or adenine bases in the genome [65,66,67]. Two classes of DNA base editors have been described: cytosine base editors (CBEs) that convert a C•G base pair into a T•A base pair, and adenine base editors (ABEs) that convert an A•T base pair to a G•C base pair. Collectively, CBEs and ABEs can mediate all four possible transition mutations (C to T, A to G, T to C, and G to A) [66,68,69]. These base editors are considered effective genome engineering tools to potentially treat human genetic diseases, two-thirds of which are due to single-base alterations [70]. However, the applications using the base editor technique have several limitations, such as (i) the identification and isolation of cell populations that have been successfully edited, (ii) the generation of some off-target effects that occur when additional cytosines that are proximal to the target base get edited, and (iii) the location of the PAM sequence for the dCas9, which limits editing efficiency, as reported in plants [71,72].

##### FokI Endonuclease Fused to dCas9

To sum up the advantages of all possible gene-editing systems, researchers have postulated that using dCas9 as a binding module and fusing it to FokI will generate a new gene-editing platform with flexible RNA-guided specificity and controlled nuclease activity by FokI dimerization [22,23,24,25,26]. This new tool is referred to as FokI–dCas9 (fdCas9) or RNA-guided FokI nuclease (RFN) and was first introduced and validated in 2014 by Tsai and Guilinger and their colleagues [22,23]. fdCas9 has indeed reported almost complete elimination of any off-target effects resulting from using WT Cas9, monomeric, and dimeric sgRNA–nickase systems (Table 2) [23]. Additional work has also validated fdCas9 activity on human genes with known off-target sites by gene-editing detection techniques that include the T7 endonuclease I (T7EI) assay and sequence alignment [22,23,24,25,26]. Similarly, Fisicaro and his colleagues used unbiased whole genome sequencing and optimized bioinformatic analysis from pigs to further prove the high efficiency and specificity of fdCas9 [73].

## 3. Engineering FokI–dCas9

To date, five constructs of fdCas9 have been engineered and published, aiming to optimize the activity, specificity, delivery, and feasibility of the system in order to encourage its usage (Table 3). The varying parameters of compositions and characteristics of the designed constructs include the following:

### 3.1. FokI Fusion to dCas9

Previously published work from our group and others attempted FokI fusion to the C-terminal of dCas9 (dCas9-FokI) to mimic ZFN and TALEN architecture, but no editing activity was detected [22,23,24]. Active fdCas9 architecture was found to be through N-terminal FokI fusion (FokI–dCas9), positioning the FokI domain away from the PAM-interacting domain and closer to the cleavage site (Figure 1, Table 1) [22,23,24]. Presuming this effect is due to structural hindrance, a C-terminal fusion would work if the polypeptide linker between dCas9 and FokI is longer and more flexible in order to span the Cas9/sgRNA complex to reach the spacer site. However, even with such an alteration, the activity of the N-terminally fused FokI was superior [22,23,24].

### 3.2. Nuclear Localization Sequence (NLS)

The functional constructs required the presence of at least one NLS upstream of the dCas9 [22,23,24,25,26]. Other constructs were engineered with multiple NLS copies at different terminus endings or within the domains to enhance nuclear delivery and the activity of fdCas9 [23,24,25,26]. fdCas9 editing efficiency has been shown to be enhanced with the addition of 3× NLS at the C-terminal, in a similar way that was used to enhance editing activity using Cre and FLPe recombinases (Table 3) [24,74]. Increasing the copies of the N-terminal NLS has been shown to increase cellular penetration and nuclear delivery of WT SpCas9 due to the positive charge of the sequence. However, all fdCas9 constructs were engineered with only one N-terminal NLS (Figure 2) [22,23,24,25,26,75]. Thus, it could be worth increasing the N-terminal NLS copy number to enhance dCas9 nuclear delivery and activity.

### 3.3. Linker

Polypeptide linkers of different lengths and amino acid compositions were tested to amend the flexibility and accessibility of the system to the genomic sites for DNA cleavage (Table 3) [23]. Editing activity was resilient to changes in the linker between the N-terminal NLS and the FokI [23]. However, significant changes in activity were detected upon alterations to the linker between the FokI and dCas9 [23]. Tsai et al. used the GlyGlyGlyGlyGlySer (GGGGS) amino acid linker, which showed a range of editing activity between 3% to 40% on various human genes [22]. In a subsequent work published by Guilinger et al., the authors tested 17 linkers with different lengths and amino acid compositions and reported varying editing activities with each linker, with the threshold activity observed with the 16-residue-long “XTEN” linker [23], similar to the linker used in our generated construct [23,24]. Later, Havlicek et al. showed that the highest editing activity was detected using the long and flexible 25-residue-long (GGGGS)_5_ linker, even when compared to previously reported GGGGS and XTEN linkers [26]. Although the shorter GGGGS linker has induced similar editing activity with a spacer distance of 14 to 18 bp, an advantage of using the (GGGGS)_5_ linker is increasing activities at the spacer length to up to 26, 29, 37, 40, and/or 41 bp apart [26].

## 4. Principles of Gene Editing Using FokI–dCas9

In order to induce the desired mutation in the target sequence using fdCas9, six distinct requirements have to be fulfilled: (i) the design of two sgRNA targeting sites flanking the gene of interest, (ii) the right spacer distance between the two targets, (iii) the two PAM sequences, (iv) PAMs must be on opposite strands, facing outwards, (v) the expression and localization of two fdCas9/sgRNA complexes onto the DNA, and (vi) the obligate dimerization of the two FokI domains (Table 2) [22,23,24,25,26]. fdCas9 is a very stringent system, and failure in fulfilling any of the listed requirements will completely abolish the DNA cleavage activity. For this reason, all published studies have noted the decreased activity of fdCas9 compared to WT Cas9 and comparable activity to paired-sgRNA nickases (Table 2) [22,23,24,25,26]. Nevertheless, all fdCas9 system requirements have resulted in significantly increased gene editing specificity, minimizing any off-target mutations that have been reported from the other gene-editing platforms [22,23,24,25,26].

To fix this dilemma of having a highly specific yet inefficient gene-editing tool, more work on understanding the precise molecular mechanism of this system will highly influence the advance of this technique. Highlighted below is an overview of the current understanding of the principles and limitations of the fdCas9 system:

### 4.1. sgRNA Design

Published studies on fdCas9 have shown a biased activity towards paired sgRNA designed on opposite strands but positioned in a PAM-out orientation, away from the target spacer sequence (Table 2) [22,23,24,25,26]. No gene-editing activity was noticed when sgRNAs were located in a PAM-in orientation or when two sgRNAs were targeting only one DNA strand [22,23,24,25,26]. A similar observation was reported when using paired nickases (D10 or H840) to induce DSBs [59]. This suggests the effect of the RNPCs’ spatial recruitment on the target DNA, facilitating strand separation to induce the DSB by fdCas9 or nickases [59].

The proper sgRNA design will highly affect the efficacy of the editing system since it determines the activity and specificity of target recognition and, thus, reduces off-target mutations. Many factors contribute to successful sgRNA engineering, including GC content of 40% to 60% and the crRNA length, which controls the level of mismatch tolerance [76]. Although the general protocol of using the CRISPR/Cas9 system has utilized 20-bp-long crRNA, a 5′ end shortening of the crRNA to 17 or 18 bp has shown a significant reduction of off-target site mutations without the loss of on-target editing activity using WT Cas9 [16]. However, this was less effective when using truncated sgRNA (tru-sgRNA) with fdCas9 [77]. Minimizing the crRNA length to 17 and 18 bp decreased fdCas9 endonuclease activity drastically, yet 19-bp-long tru-sgRNAs showed comparable on-target gene editing activity to full 20-bp-long gRNAs [77]. Although negligible off-target mutation occurred at low rates using fdCas9, its usage with a 19-bp tru-sgRNA diminished those rates to almost nonexistent [77].

### 4.2. PAM Sequences and fdCas9 Variants

The fdCas9 system will only be active in the presence of two PAM sequences that are located at opposite stands and are 15 to 26 bp apart, which makes the possibility of finding those requirements around the gene of interest lower. To expand the range of PAM sequence recognition, other CRISPR systems and Cas9 orthologs can be used. All previously described fdCas9 constructs have used the spCas9 variant recognizing NGG PAM. The recent identification of other Cas9 orthologs, such as *Staphylococcus aureus* Cas9 (SaCas9), which recognizes the NNGRRT PAM sequences, has been successfully used as a FokI-mediated nuclease system in its dead SaCas9 (dSaCas9) form [26]. Although editing efficiency using pairs of FokI–dSaCas9 has shown a lower activity to that of pairs of FokI–dSpCas9, Havlicek’s group has reported that heterodimerization using both FokI–dSaCas9 and FokI–dSpCas9 orthologs is possible, resulting in a fdCas9 system with intermediate gene editing activity [26]. In addition, chimeric SaCas9 variants were engineered to expand on the PAM recognition sequences [78]. This can provide an expanded range of dSaCas9 variants that can be potentially engineered to provide an array of available RNF systems [78]. However, the delivery of multiple dCas9 variants will need to be addressed and optimized in this case.

### 4.3. Obligate FokI Dimerization

The expression of only a single fdCas9/sgRNA complex reported no gene-editing activity [22,23,24,25,26]. The nuclease was only activated when pairs of heterodimer fdCas9/sgRNA were expressed [22,23,24,25,26]. It has been previously reported that the optimal spacer length for ZFNs is 5 to 7 bp; for TALENs, it is 14 to 16 bp [12,32]. Our data demonstrated that fdCas9 exhibited DSB cleavage with spacer lengths ranging from 15 to 39 nucleotides [24], consistent with results obtained by other studies using various constructs of fdCas9 bearing spacer ranges of 16 to 18 bp and 25 to 26 bp [22,23,25]. The strict spacing between the sgRNA pair was not seen when using sgRNA pairs in nickases but was seen in FokI-mediated ZFNs and TALENs [12,32,59]. This suggests that this strict spacer requirement is optimal for FokI domain recruitment, dimerization, and DNA cleavage. Since FokI nuclease is a type II bipartite endonuclease, it cleaves DNA upon recognition of two DNA sites that are 10 bp apart [32,79]. This allows the formation of a DNA loop to facilitate the recruitment of the two FokI catalytic domains into close proximity for dimerization and DNA cleavage [79].

Although FokI dimerization has remarkably resulted in diminishing any off-target effect that could arise from using WT or nickases Cas9, dimerization is limited by a defined spacer distance and PAM location, reducing the chances of finding candidate genes [80]. An additional limitation is the fact that two fdCas9/sgRNA complexes have to be expressed and present at the target site for successful editing, which means that the number of copies to be synthesized is doubled as compared to WT Cas9 [80]. On the other hand, the advantage of using FokI with Cas protein was seen in the CRISPR/Cas type I system [81]. Generally, type I systems are difficult to use as gene-editing tools because they require the assembly of more than one Cas protein to form the active endonuclease system, as in the case of the Cascade-Cas3 system involving five different Cas proteins [81]. Nonetheless, FokI fusion to Cas8, which is a part of the EcoCascade, showed a significant gene-editing efficiency of greater than 50% [81]. This was the first class I CRISPR gene editing activity to be reported and used in human cell lines, and this was only achieved by the activity of the fused FokI domain [81].

### 4.4. fdCas9 System Delivery

CRISPR/Cas9 system cellular delivery is one of the main limitations to its wide range of applications [82]. Lentiviral vectors are commonly used for specific ex-vivo and in-vivo cellular delivery [76]. However, some disadvantages include the toxicity and immunogenic effects, as well as the limited DNA capacity of about 4.7 kbps that restrains their use [76]. Nonviral ex-vivo delivery is also possible, including techniques such as microinjection, lipofection, electroporation, nanoparticles, cationic peptides, and extracellular vesicles [83]. Genome-editing system components can be delivered in the form of DNA/plasmids encoding both the Cas complex and the sgRNA transcript [51]. Drawbacks of DNA delivery, however, are the prolonged and continuous expression of the editing system (which was found to induce toxicity in bacteria), the possibility of host genome integration (which can increase genomic instability), and the generation of off-target mutations [76,84,85]. To resolve these issues, new strategies of delivering another CRISPR system directed at terminating the editing system’s expression has been used [86]. A point to consider, however, is that an excessive amount of DNA delivery is highly toxic to cells, and so, other means of delivering the system’s components are favored [87]. Options include delivery using mRNA transcripts, transcribed sgRNA, or as an active RNPC [88]. This can be chosen depending on many characteristics, including the cost, stability through the delivery process, efficiency, rapidity and duration of gene-editing, toxicity, off-target effects, and immunogenicity [83].

In context of fdCas9, ex-vivo delivery methods tested include lipofection, electroporation nucleofection, and microinjection of fertilized mice oocytes [23,24,25,89,90,91]. Different delivery methods have been tested to enhance editing activity, such as fdCas9/sgRNA as mRNA transcripts and sgRNA as crRNA and tracrRNA separately, all of which resulted in comparable gene-editing activities as DNA delivery [91,92,93]. So far, no published work has been done to test the effect of fdCas9 RNPC delivery since it has not been purified and it is not commercially available yet. Lentiviral delivery has not yet been used either as it requires further optimization to fit the large size of the fdCas9 construct and dual sgRNAs [80]. There are some solutions to solving this issue, including splitting dCas9 domains into multiple deliveries [94], minimizing the size of the constructs and forming truncated dCas9 [95], or replacing dSpCas9 with the smaller dSaCas9 [89,94,95].

## 5. Applications of FokI–dCas9

Along with fdCas9 activity validation using various human cell lines, fdCas9 was also tested ex-vivo using mice and pig models [25,90,91,92,93]. Nakagawa’s group compared the efficiency of gene editing in fertilized mice oocytes induced by single Cas9/sgRNA, double nickase/sgRNAs, and double fdCas9/sgRNAs and reported that fdCas9 provided a balanced ratio of mutations to mice birth rates [25]. In addition, an all-in-one construct encoding for fdCas9 and two sets of sgRNA was engineered to target two different genes for double-knockdown applications (Figure 2) [25]. The construct has shown that fdCas9 multiplexing to two different loci that are 96.1 kb distance apart facilitated double-gene-knockout generation in mice at a successful rate of more than 50% and as a single knockout, of nearly 70% [25]. fdCas9 was also used successfully to knock-in a transgene into a pig genome, where one allele replacement was induced by HDR and the other allele was inactivated by NHEJ repair [90]. Moreover, fdCas9 was also successfully used in a Phenylketonuria cell-line model to correct the defective gene encoding phenylalanine hydroxylase [95]. It is clear that this and other studies provide promising applications for fdCas9 as a therapeutic genome engineering tool.

## 6. Conclusions

fdCas9 is a robust and efficient system for genome engineering applications, especially in genomic medicine. It has a great advantage in overcoming off-target effects associated with other genome engineering tools, including ZFNs, TALENs, and CRISPR/Cas systems.

We conclude that all the engineered fdCas9 variants with N-terminal FokI fusion induce significant gene-editing activities and specificities, as tested on various gene targets, when compared to other genome-engineering tools [22,23,24,25,26]. Like ZFNs and TALENs, the fdCas9 system is active only as a heterodimer, requiring (i) the simultaneous binding of two fdCas9/sgRNAs monomers at adjacent target sites in a PAM-out orientation, and (ii) a specific spacer distance separating the two binding sites of the two sgRNAs. This spacer distancing is considered a major factor that limits genome editing with fdCas9. We believe that redesigning and engineering various peptide linkers between the FokI endonuclease domain and dCas9 can improve the spacing distance flexibility for fdCas9 dimerization and, thus, provide better gene editing activity.

## Figures and Tables

**Figure 1 cells-09-02518-f001:**
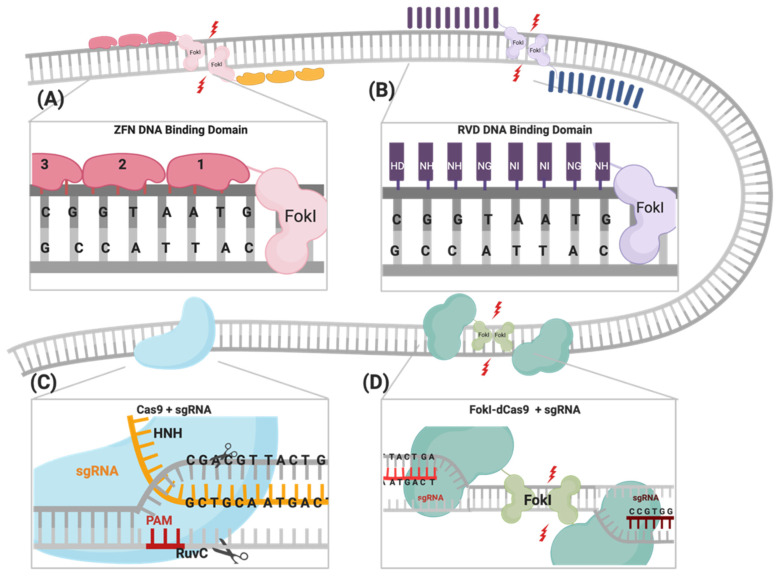
Illustrations of different genome engineering tools. (**A**) Zinc fingers nucleases (ZFNs) bind to the target sequence through 3-4 zinc finger proteins, and (**B**) transcription activator-like effector nucleases (TALENs) through repeat variable diresidues (RVDs), both of which inducing double-stranded breaks (DSBs) through heterodimerization and the catalytic activity of the fused FokI. (**C**) CRISPR–Cas9 binds to the DNA target through PAM sequences and the complementary sgRNA site, where the catalytic activity is mediated through HNH and RuvC domains. (**D**) FokI–dCas9 (inactive “dead” form of Cas9) binds to the target site through two sgRNAs that are positioned in a PAM-out orientation, and the catalytic activity is mediated through the dimerization of fused FokI endonuclease.

**Figure 2 cells-09-02518-f002:**
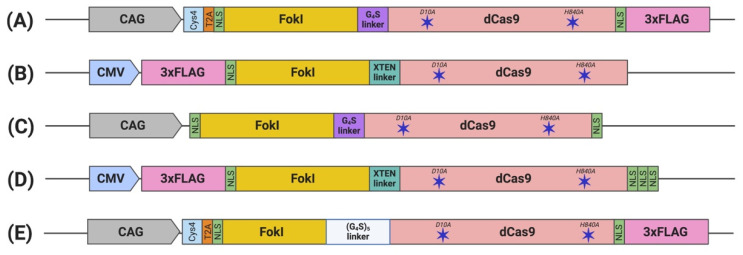
Schematic illustration of engineered FokI–dCas9 (fdCas9) constructs. Constructs are generated by (**A**) Tsai et al., (**B**) Guilinger et al., (**C**) Nakagawa et al., (**D**) Aouida et al., and (**E**) Havlicek et al. fdCas9 expression is driven by a strong constitutive human promoter (CMV and CAG). FokI endonuclease is fused to the N-terminal of the dead Cas9 (dCas9) [22,23,24,25,26].

**Table 1 cells-09-02518-t001:** Comparison between different gene-editing tools.

	Zinc Finger Nucleases (ZFNs)	Transcription Activator-Like Effector Nucleases (TALENs)	CRISPR/Cas Systems	
			CRISPR Associated Endonuclease (Cas9)	FokI Dead Cas9 Endonuclease (FokI–dCas9)
DNA catalytic domain	FokI	FokI	RuvC and HNH	FokI
DNA recognition	DNA: Protein		DNA: RNA	
unit of target recognition	Pairs of ZFNs (via the ZF motifs)	Pairs of TALENs (via RVD tandem repeat).	One 17–20 bp sgRNA	Pairs of 19–20 bp sgRNAs
Recognized target size	Recognizes 18-24 bp	Recognizes 30–40 bp.	Recognizes NGG PAM sequence + 17–20 bp	Recognizes two NGG PAM sequences + 38–40 bp
Specificity	Tolerates few positional mismatches		Tolerates both positional and multiple consecutive mismatches	Enhanced specificity due to the dual sgRNAs requirement of
Spacer size	5–7 bp	14–16 bp	No spacer required	13–18 bp and/or 26 bp
Ease of delivery	Limited delivery due to the difficulty of linking ZF modules	Difficult delivery due to cDNA size and extensive TALEs repeats	Easily delivered using standard delivery and cloning techniques	Harder to deliver due to increased size of construct and added components
Limitation	Off-target effects Limited delivery due to size constraints	Off-target effects Expensive	Off-target effects due to mismatch tolerance PAM sequence availability	Difficult to deliver A strict system with many obligatory requirements
Multiplexing	Difficult		Easy, can form multiplexes directed to multiple genes	

**Table 2 cells-09-02518-t002:** Comparison between genomic editing activities of Nickases versus FokI–dCas9.

	Nickases (Pairs of D10 or H840)		FokI-dCas9	
Structure	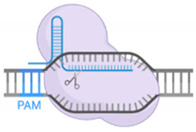	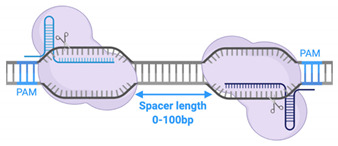	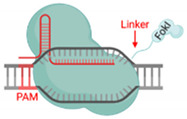	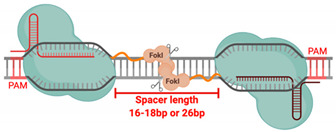
Existing forms	Monomers	Dimers	Monomers	Dimers
Cleavage domain	HNH or RuvC	Pairs of HNH or pairs of RuvC	FokI *(But inactive)*	Pairs of FokI
Obligate dimerization	-	No	-	Yes
Spacer length	-	Up to 100bp	-	16-18bp or 26bp (depending on variant used)
Target size	17-20bp	34-40bp + spacer length	19-20bp	38-40bp + spacer length
Linker	-	-	-	Required to link the FokI domain
Type of DNA damage	Single strand nicks	staggered double strand break	No damage induced	Double strands break
Type of mutations	Can induce point mutations	Additions or deletions of >2 bps	Non-mutagenic	Additions or deletions of >2 bps
Off-target effect (Compared to WT Cas9)	Low-moderate	Low	Nearly non-existing	Rare

Illustration figures created by BioRender.com.

**Table 3 cells-09-02518-t003:** Comparison between different fdCas9 constructs available in the literature.

	sgRNA Delivery Method	Linker	Gene Editing Activity of fdCas9			Optimal Spacer Distance	Off-Targets	
			Compared to Negative Control	Compared to Single WT Cas9	Compared to Paired Casas9 nickases		Genes Tested	Activity
Tsai (2014) [22] (Figure 2A)	Csy4-based dual sgRNA expression system	GGGGS linker	3–40%	Differences varied depending on gene tested	Similar to fdCas9	13–18 bp	VEGFA	Indistinguishable off-target mutation
Guilinger (2014) [23] (Figure 2B)	dual sgRNA expression plasmid	17 linkers tested; best activity using XTEN linker	GFP disruption: 10% reported by flow cytometry and 20% by T7EI	eGFP disruption: 25% reported by flow cytometry and 2/3 of fdCas9 activity by T7EI	eGFP disruption: 15% reported by flow cytometry and similar activity to fdCas9 by T7EI	15 and 25 bp	AAVS1, CLTA, EMX1, HBB, VEGFA	Fold increase of on/off-target editing: 140 compared to WT Cas9 1.3–8.8 compared to nickases
			Average of 14.9% on human genes	Average of 28.2% on human genes	Average of 20.6% on human genes			
Nakagawa (2015) [25] (Figure 2C)	All-in-one construct, included in the fdCas9 plasmid	GGGGS linker	Tested on mice fertilized oocyte			13–18 bp	Top three candidates of used sgRNA	No off-target editing was reported
			Average of 49%, and moderate birth rate	Average of 90%, and a low birth rate	Average of 2.9%, and a high birth rate			
Aouida (2015) [24] (Figure 2D)	sgRNA expressing DNA fragments	XTEN linker	eGFP disruption: 5% reported by flow cytometry	eGFP disruption: 12.3% reported by flow cytometry	eGFP disruption: 1% reported by flow cytometry	15–39 bp	CCR5, AAVS1, EMX1, HBB	Only WT Cas9 showed 25–30% off-target editing
Havlicek (2017) [26] (Figure 2E)	Csy4-based multiple sgRNA expression systems	(GGGGS)_5_	Average of about 30% human gene editing	Cas9 orthologs including SpCas9-HF1 and eSpCas9 showed higher activity	*Not tested*	13–18 and 26 bp	CLTA, EMX1, VEGFA	fdCas9 outperform limited off-target editing compared to all Cas9 orthologs tested

GFP- green fluorescence protein, AAVS1- Adeno-Associated Virus Integration Site 1, CLTA- Clathrin Light Chain A, EMX- Homeobox protein 1, HBB- Hemoglobin Subunit Beta, VEGFA- Vascular Endothelial Growth Factor A, CCR5- chemokine receptor 5.

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
