# Peer review of "CRISPR FokI Dead Cas9 System: Principles and Applications in Genome Engineering"

_cells, 2020, doi:10.3390/cells9112518_

Round 1
Reviewer 1 Report
The review article by Saifaldeen et al. summarize the role of FokI-dCas9 fusion proteins as tools for genome engineering. First, the authors describe the main approaches to exploit the non-specific FokI nuclease domain in fusion with different DNA binding modules for applications in gene editing. They illustrate advantages and limitations of each system and compare it to their favored FokI-dCas9 fusion protein. Finally, the authors address different requirements to optimize the FokI-dCas9 fusions for gene editing applications.
Major points
- While the review is generally well written, it reads like a text copied from a Ph.D. thesis with undeleted additional citations and an incomprehensible labeling of sections. (e.g. page 13: section 4.1?)
Please reorganize. - The authors should include and discuss the use of obligate heterodimer variants of the FokI in Cas9 fusions as suggested by Tsai and his colleagues and Guilinger and his colleagues.
Minor points
- Page 1, line 48: “One year later, …”. ???
- Page 2, line 50: Here the authors should quote Tsai and Guilinger [18,19]. These groups introduces the FokI-DCas9 system.
- Page 2, line 64: “zinc ion” not “zinc molecule”
- Page 2, figure 1: ZFNs and TALENS are heterodimers, please use different colors for the DNA binding modules.
- Page 3, line 87: “fused FokI” it is the catalytic domain of FokI !
- Page 3, line 128: double citations
- Page 4, line 132: Please specify the cas9 protein e.g. spCas9
- Page 5, table 1, title: tools
- Page 6, line 5: effector not “affecter”
- Page 6, last paragraph: text is corrupt
- Page 12: double citations
- Page 12: “section 4.1” ????
Reviewer 2 Report
Line 35 …… which is flanked by complementary overhangs to allow gene displacement using the cellular homologous recombination
- …… which is flanked by complementary overhangs homology arms to allow gene replacement using the cellular homologous recombination?
Line 37…. HDR mechanism is considered to be a slow-occurring process and with low success rates, 37 but is favored for precise gene knock-in and knock-out applications [4].
- The reference is focused in CAs9 no in the process of HDR at all. Pleas change it.
Line 40…. resulting in shifting the reading frame [5].
- There are more recent references focused in NHEJ issue
Line 46 have been utilized with great success [6].
- The reference is no appropriate
Line 48 ……One year later, a new derivative of the CRISPR genome
- I don’t understand one year later refer to what??’
Line 51 In this review, we will compare all the above mentioned gene engineering tools to fdCas9
- But the review start talking about ZFN and Talen, please include those systems in the introduction or eliminated from the review
Line 112 Although ZFNs and TALENs have high genome editing efficacy, they were not broadly adapted in the scientific research community due to the complexity of protein engineering required skills [34].
- I´m completely disagree with this affirmation, almost Clinical trial already in clinic are based on TALEN and ZFN; pleas rephrase this affirmation.
Line 122 during infections by bacteriophages and is now repurposed as a powerful genome engineering tool
- The reference 40 is no appropriate
Line 137 One of the important advantages of the CRISPR/Cas9 system compared with other genome engineering tools, such as ZFNs and TALENs, is that multiple sgRNAs can be introduced simultaneously to create multiplex targeting mutations.
- This not totally true, TALEN and also be multiplexed., pleas refer to following reference:
Mol Ther Nucleic Acids. 2017 Dec 15;9:312-321. doi: 10.1016/j.omtn.2017.10.005.
Line 139 However, there are few limitations of using CRIPCR/Cas9 system in genome engineering that include the PAM sequence availability at target sites, which is estimated to be one every 8bp in the human genome [9].
- I’m not sure about this assertion…..
Line 150 or RuvC nuclease activity being abolished by the variant D10A and H840A, respectively[45,49].
- These two refences do no refer to nickase systems….
There is no line number from page 6.
System delivery:
The author do not clarify if they talk about ex vivo or in vivo delivery…., from the lecture, I deduce that they are talking about the in vivo approach, in any case, the ex vivo delivery is the ,most promising approach, for the moments. However RNP method is mentioned, this method as far as I know is exvivo based method.
This and many many other imprecision’s should be rewrited and clarified before accepting for publication
Reviewer 3 Report
Review of Cells-963351
CRISPR FokI Dead Cas9 System: Principles and Applications in Genome Engineering
General comments: This review article summarizes commonly used nuclease systems for gene editing, with a particular focus on the FokI-based dead Cas9 system that was developed in and around 2014-2015. The FokI-dCas9 system has not seen widespread adoption by the gene editing community but nonetheless has a number of interesting differences from the more commonly used Cas9 single and dual nuclease systems. In general, I found the manuscript difficult to read and it would benefit tremendously from language editing. I think the general conclusion that the FokI-dCas9 system has reduced (or abolished, to use the author’s words) off target toxicity is naïve and potentially a dangerous message for the gene editing community. I strongly suggest that this conclusion be re-stated to accurately reflect that estimates of off target toxicity for any gene editing nuclease are problematic as they are dependent on the methodology of assessing off-target effects, cell line/organism, delivery method and nuclease expression levels, and a host of other factors that determine nuclease efficiency in cells.
Specific comments:
- FokI should not be italicized following the published conventions for nomenclature of restriction enzymes and derivatives (https://academic.oup.com/nar/article/31/7/1805/1193058)
- Consistency between sgRNA and gRNA – pick one (preferably sgRNA) and use throughout manuscript.
- The manuscript needs extensive editing. I have noted some examples here. The authors are strongly encouraged to use a manuscript editing service.
- Page 1, line 10. First sentence of abstract is misleading. How can an enzyme system be superior yet have high frequency of off-target activity? This needs re-phrasing.
- Page 1, line 19. Similarly, is it true that all FokI dCas9 variants are superior platforms for gene editing in eukaryotic systems?
- Page 1, line 34: HDR is a mostly error free process, it is incorrect to claim it is completely error free
- Page 1, line 37. “slow-occurring process” is awkward. Perhaps low efficiency is better?
- Page 1, line 39. NHEJ does not involve the addition or deletion of short nucleotide sequences. It is more appropriate to say that an outcome of NHEJ repair is the deletion or insertion of nucleotides at the break site. Also, there is no distinction made between different NHEJ pathways that operate on DSBs.
- Page 1, line 40. The term “higher eukaryotic” is meaningless. Are the authors confident that NHEJ is the dominant pathway in so-called higher eukaryotes? It is the dominant pathway in many cell-line based systems.
- Page 1, line 41: I did not see the claim in reference 4 study that NHEJ is ideal for gene manipulation that aims for gain or loss of gene function
- Page 1, line 48. “One year later” is confusing. What year?
- Page 2, figure 1: figure resolution is very poor, and though the caption refers to A, B, C, and D, these are not labeled in the figure. Repeat variable di-residues mentioned in caption, but should also include RVD acronym for clarity, as it is what is used in the figure itself.
- Page 3, lines 101-102: “TALENs are distinct from ZFNs as they bind in a ratio of 1:1 protein to DNA, resulting in a broader target site selection” unclear how the binding ratio is relevant in broader target site selection. Clarify.
- Page 3, lines 100-103: some readers may not be familiar with the term spacer. Should define that this is the distance on the DNA target between the two binding sites.
- Page 3, line 106: “the non-specific cuts raise a concern” what non-specific cuts? How are cuts non-specific if targeted to a sequence by the TALE DNA binding domain? Clarify.
- Page 3, line 110. The heading “genome editing tools utilizing nucleotides as DNA binding motif” is very confusing. Cas9 uses both the sgRNA and specific amino acids to contact DNA (at the PAM site). The way the heading is written makes it sound like Cas9 only uses the sgRNA to interact with the target site, which is not true.
- Page 3, line 121. “..against pathogens during infections by bacteriophages…” is confusing. Delete pathogens during infection.
- Page 4, line 132. The PAM is specific to the Cas9 protein. The NGG required is for the S.pyogenes Cas9. Please make that clear.
- Page 4, line 135. The sentence “Typically in higher eukaryotes…” is not needed here. This section is not about NHEJ repair.
- Page 4, line 137. Multiplexing is not unique to Cas9 systems and I’m not sure this is major advantage over TALENs and ZFNs. Rather, I would say the most significant advantage is ease of programming targeting specificity by simply changing the sgRNA sequence.
- Page 4, line 144. crRNA?? What is this? Define.
- Page 5: table requires significant redesign, spacing is inconsistent and random, words in adjacent columns too close and difficult to distinguish which column they belong to. Caption should say tools, not tool.
**** annoyingly, the line numbering stops after page 5 ****
- Page 6, first paragraph. dCas9 is still toxic in bacteria. https://www.nature.com/articles/s41467-018-04209-5
- Page 6, second paragraph. Here, or somewhere in the manuscript, the authors should take time to read, mention, and cite the work of Stephen Halford on how FokI can cleave half-sites, even with the engineered obligate heterodimers.
- Page 6, paragraph 3: incorrect labeling of headers, dCas9 fusion with FokI endonuclease should be number 4, no further sections numbered.
- Page 7, end of first paragraph. Ref 64 does not compare editing activity of fdCas9 with other nucleases, it only examines editing by fdCas9. This sentence is misleading as written.
- Page 7, paragraph 2: “mutation rates reported could be resulted from random point mutations, that may be insufficient to induce desired genomic changes”. This is confusing. Clarify.
- Page 7, third paragraph. All of the factors listed here are also disadvantages of the fdCas9 system compared to other Cas9 nucleases.
- Page 8: table 2 figures are very small, caption is above table instead of below as in tables 1 and 3. Fix spacing between words.
- Page 10: table 3 also requires a significant redesign to improve clarity.
- Page 12, heading of first paragraph does not make sense. “FokI terminus fusions to dCas9”. What terminus?
- Page 12, paragraph 1: section 4.3.3 does not exist.
- Page 13, paragraph 2: “This shows that mutations generated by fdCas9 favors NHEJ repair machinery” an odd claim to make, as NHEJ is usually the quicker repair pathway for most gene editing techniques.
- Page 13, paragraph 3: “heterodimerization using both dSaCas9 and dSpCas9 is possible” this would introduce another limitation not mentioned in terms of delivery, as now two distinct Cas9 fusion proteins must be delivered to the cell, as opposed to one.
- Page 14, conclusions. Again, I think it is naïve and dangerous to state that “Due to its high specificity, almost all unexpected and undesired mutations are abolished”. The literature does not support this statement – there are few examples of comprehensive, unbiased genome wide screens of fdCas9 activity.
Other minor edits
- Page 1, line 30: should be defective, not defected
- Page 1, line 31: should be engineered, not engineering
- Page 1, line 31: “would stimulate” should be “stimulates”
- Page 1, line 32: “resulting” should be “results”
- Page 1, line 33 and 34: “the” before homology directed repair and non-homologous end joining is unnecessary
- Page 1, line 46: “nucleases-based” should be “nuclease-based”
- Page 1, line 47: “creates some undesired off-target” > “creates undesired off-target”
- Page 2, line 49: specify “a catalytically inactive form of Cas9”
- Page 2, line 78: to the of human genome
- Page 3, line 87: should note that the fused FokI nuclease in fdCas9 also requires dimerization for functional activity
- Page 3, line 93: a bacterial pathogen of plants.
- Page 3, line 97: “These repeats have a polymorphism that is restricted to only a pair of residues known as repeat variable di-residues (RVD),…”
- Page 3, line 98: located at the positions 12 and 13
- Page 3, line 99: reference 19 formatting. Also C-terminal should be C-terminus.
- Page 3, line 100: “was emerged” > “was created”, or just “emerged”
- Page 3, line 115: “system have” should be “systems have” or “system has”
- Page 3, line 125: single guide RNA, not single guided RNA
- Page 3, line 125: references 42 and 43 should be in same brackets; formatting
- Page 3, line 125: “The sgRNA molecule”
- Page 3, line 127: “The sgRNA is” or “sgRNAs are”
- Page 3, line 128: reference formatting
- Page 4, line 132: different Cas9 orthologs recognize different PAM sequences; not mentioned until much later in paper
- Page 4, line 133: and the sgRNA
- Page 4, line 138-139: “multiple sgRNAs can be introduced simultaneously to create multiplex targeting mutations” while this is an important and useful feature, the most important advantage of Cas9 over other methods is the ease and simplicity of changing the target sequence
- Page 4, line 140: CRIPCR > CRISPR
- Page 4, line 141: should be a space between number 8 and unit bp
- Page 4, line 143: “which occurs” > “that occurs”
- Page 4, line 148: “resulted from” > resulting from the”
- Page 6, paragraph 1: “is formed by the point mutation of both the endonuclease domains HNH and RuvC” should reword this sentence to improve clarity
- Page 6, paragraph 1: “affecter proteins” should be “effector proteins”
- Page 6, paragraph 1: “via fusing” should be “via fusion”
- Page 6, paragraph 1: “epigenetics modification” should be “epigenetic modification”
- Page 6, paragraph 1: “and many other” should be “and many others”
- Page 6, paragraph 2: should be “CBEs that convert” and “ABEs that convert”
- Page 6, paragraph 3: formatting errors in references
- Page 6, paragraph 3: “subsequently improve the specificity of the editing as compare to” > “subsequently improves the specificity of the editing as compared to”
- Page 6, paragraph 3: I wouldn't say all the Cas9 derivatives, since you have not examined every single Cas9 derivative in this review. perhaps the previously mentioned Cas9 derivatives.
- Page 6, paragraph 4: “by Tsai and his colleagues and Guilinger and his colleagues” Provide in text citation.
- Page 6, paragraph 4: specify if it is wild-type Cas9
- Page 7, paragraph 2: “wild type Cas9” > “wild-type Cas9”
- Page 9, paragraph 4: “encourage the use of this a new genome”
- Page 9, figure 2: poor resolution
- Page 12, paragraph 1: “C terminal” > “C-terminal”
- Page 12, paragraph 2: “The functional constructs required the presence of at least one NLS upstream and one downstream of the fdCas9 to ensure nuclear delivery of the fdCas9 complex, to enhance nuclear delivery, as compared to only a single NLS either upstream or downstream some constructs were generated with multiple NLSs at different terminal locations and within the domains to optimize the activity of fdCas9” severe run on sentence, unclear. Rewrite to improve clarity
- Page 12, paragraph 2: several reference formatting errors
- Page 12, paragraph 2: “plays an essential role in producing functionally active fdCas9 in the nucleus” I don't know if this is strictly accurate in terms of "producing functionally active fdCas9”; it is just required to localize the fdCas9 to the nucleus, as it would not exert its editing effects in the cytoplasm?
- Page 12, paragraph 4: “fdCas9 provides a balanced ratio” > “fdCas9 provided a balanced ratio”
- Page 12, paragraph 4: “In addition, all-in-one construct” > “In addition, an all-in-one construct”
- Page 12, paragraph 4: “96.1-kb” no dash between number and units, just a space
- Page 12, paragraph 4: “fdcas9” > “fdCas9”
- Page 13, paragraph 1: sections 4.2 and 4.3 do not exist
- Page 13, paragraph 1: “using fdCas9 system for” should be either “using the fdCas9 system for” or “using fdCas9 for”
- Page 13, paragraph 4: section 4.1 does not exist
- Page 13, paragraph 4: “editing efficiency to more than 50%.” > “editing efficiency greater than 50%.”
- Page 14, paragraph 1: “off target” > “off-target”
Round 2
Reviewer 1 Report
The manuscript is now suitable for publication.
Reviewer 2 Report
from my point of view, the authors, adressed all the suggested points
and the current version of the review is suitable for publication.